# Endoscopic Ultrasound-Guided Fine-Needle Biopsy Using 22G Franseen Needles without Rapid On-Site Evaluation for Diagnosis of Intraabdominal Masses

**DOI:** 10.3390/jcm11041051

**Published:** 2022-02-17

**Authors:** Nonthalee Pausawasdi, Kunsuda Cheirsilpa, Wipapat Chalermwai, Ishan Asokan, Tassanee Sriprayoon, Phunchai Charatcharoenwitthaya

**Affiliations:** 1Siriraj GI Endoscopy Center, Faculty of Medicine Siriraj Hospital, Mahidol University, Bangkok 10700, Thailand; kunsuda.cheirsilpa@hotmail.com (K.C.); tassanee2449@yahoo.com (T.S.); phunchai@yahoo.com (P.C.); 2Division of Gastroenterology, Department of Medicine, Faculty of Medicine Siriraj Hospital, Mahidol University, Bangkok 10700, Thailand; iasokan@mednet.ucla.edu; 3Division of Gastroenterology and Hepatology, Department of Medicine, Chulabhorn Hospital, Chulabhon Royal Academy, Bangkok 10210, Thailand; 4Department of Pathology, Faculty of Medicine Siriraj Hospital, Mahidol University, Bangkok 10700, Thailand; wipapat.api@mahidol.edu; 5Department of Internal Medicine, David Geffen School of Medicine, University of California Los Angeles, Los Angeles, CA 90095, USA

**Keywords:** endoscopic ultrasound, fine-needle biopsy, fine-needle aspiration, rapid on-site evaluation, histology, pancreatic cancer

## Abstract

Background: The impact of rapid on-site cytologic evaluation (ROSE) on endoscopic ultrasound-guided fine-needle biopsy (EUS-FNB) is widely debated. This study aims to assess the diagnostic performance of EUS-FNB in the absence of ROSE in abdominal masses. Methods: Patients with abdominal masses undergoing EUS-FNB using 22-gauge Franseen needles and the slow-pull technique were prospectively enrolled in this study. Macroscopic on-site evaluation (MOSE) was performed without ROSE. Results: 100 patients were recruited between 2018 and 2020. Seventy-eight patients had neoplasms, and twenty-two patients had benign diseases. Common diagnoses included pancreatic cancer (*n* = 27), mesenchymal tumors (*n* = 17), and metastatic tumors (*n* = 14). The mean mass size was 3.9 ± 2.6 cm. The median pass number was three. Eighty-nine percent had adequate specimens for histologic evaluation. Malignancy increased the odds of obtaining adequate tissue (OR 5.53, 95% CI, 1.36–22.5). For pancreatic cancer, FNB had a sensitivity of 92.3%, a specificity of 100%, a positive predictive value (PPV) of 100%, a negative predictive value (NPV) of 97%, and an AUROC of 0.96. The sensitivity, specificity, PPV, NPV, and AUROC for mesenchymal cell tumors were 100%, 95.9%, 84.2%, 100%, and 0.98, respectively. For metastatic tumors, FNB was 100% sensitive and specific, with an AUROC of 1.00. There were no procedure-related complications. Conclusions: 22-gauge Franseen needles with the slow-pull technique and MOSE without ROSE provide excellent diagnostic performances for malignant lesions. Thus, MOSE should be implemented in real-world practice, and ROSE can be obviated when EUS-FNB is employed.

## 1. Introduction

The inability to collect core tissue and inadequate sampling remain formidable limitations of endoscopic ultrasound-guided fine-needle aspiration (EUS-FNA). EUS-FNA is primarily used in the biopsy of structures adjacent to the gastrointestinal tract; however, this method has many challenges. Fine-needle biopsy (FNB) has now been engineered to overcome these limitations. One of the novel biopsy-needle designs with promising diagnostic performance is the Franseen tip design. This needle features a crown tip with three symmetrical cutting edges that enable swift core tissue procurement for histological analysis. Several studies have shown that Franseen needles provide greater than 95% tissue adequacy and an excellent overall diagnostic accuracy of more than 90% [1,2,3,4,5,6,7]. As a result, the diagnostic strategy of EUS-guided tissue acquisition has shifted to endoscopic ultrasound-guided fine-needle biopsy (EUS-FNB) for both pancreatic and non-pancreatic lesions. The aspiration techniques, the role of rapid on-site evaluation (ROSE) for cytological analysis, and macroscopic on-site evaluation (MOSE) have been investigated to refine the diagnostic performance of EUS-FNB. Previous studies used a variety of aspiration techniques, including no suction, suction with 10–20 mL of negative pressure, and the slow-pull technique [1,2,3,4,5,6,7]. The use of negative-pressure suction was associated with lower diagnostic accuracy and more blood contamination than no suction and the slow-pull technique [7]. A small retrospective study demonstrated the usefulness of MOSE in evaluating visible core tissue with fewer passes required to obtain adequate tissue sampling [8]. The role of ROSE in EUS-FNB is the current topic of investigation. A randomized trial showed that EUS-FNB without ROSE is not inferior to EUS-FNA with ROSE in diagnosing pancreatic lesions. The added benefit of EUS-FNB further hinges upon its need for fewer needle passes [9]. The use of EUS-FNB without ROSE is now expanding, but evidence to support this shift in practice is scarce. Factors affecting tissue adequacy and diagnostic performance of EUS-FNB when ROSE is not applied have yet to be explored.

This prospective study aims to prove the effectiveness and diagnostic performance of 22-gauge Franseen needles using the slow-pull aspiration technique for histological analysis with MOSE in the absence of ROSE. This study included only EUS of the upper gastrointestinal tract.

## 2. Materials and Methods

### 2.1. Study Design

This prospective study was conducted at a tertiary care center. The primary aim was to assess the diagnostic performance of 22-gauge Franseen needles for histological analysis without ROSE. The secondary aims included assessing the Franseen needle’s ability to provide adequate tissue for histological examination, exploring factors associated with tissue adequacy, and the safety profile. Informed consent was obtained from all patients before study enrollment. This study was conducted following the Declaration of Helsinki and approved by the Institutional Review Board. The Thai Clinical Trial Registration identification number documenting the study is TCTR20200823001.

### 2.2. Patient Population

Patients who underwent EUS-FNB of the upper gastrointestinal tract for intra-abdominal solid lesions were prospectively enrolled between 2018 and 2020. Inclusion criteria included (1) age > 18 years, (2) intra-abdominal solid masses detected by cross-sectional imaging, and (3) lesions accessible by EUS. Exclusion criteria included (1) cystic lesions, (2) coagulopathy (international normalized ratio, INR > 1.5), (3) thrombocytopenia (platelet count <50,000 mm^3^), (4) contraindications for conscious sedation, (5) pregnancy, and (6) failure to obtain informed consent. Patient demographic data, clinical manifestations, endosonographic findings, FNB methods, tissue handling and processing, and complications were collected and analyzed.

### 2.3. EUS-FNB Techniques

EUS was performed by an experienced endoscopist, who previously had performed more than 2000 EUS cases at a tertiary care center, using a linear array echoendoscope (GF-UC140P or GF-UC160P, Olympus Corporation, Tokyo, Japan) and a processor (Pro-Sound Alpha-10 or Pro-Sound F75, Hitachi Aloka Medical, Ltd., Tokyo, Japan). The lesions were accessed using a 22-gauge Acquire needle^®^ (Boston Scientific Corporation, Natick, MA, USA) with the slow-pull technique. MOSE was performed after each pass. If insufficient material was obtained, repeat pass attempts were made until a visible tissue of ≥4mm in length was collected.

The EUS-FNB technique includes target lesion identification, doppler evaluation, needle puncture, tissue aspiration, and specimen handling. Once the lesion was localized, a color doppler was applied to evaluate the intervening vessels. After identifying the appropriate window without intervening vessels, the stylet was slightly withdrawn to sharpen the needle. The needle was advanced into the lesion and moved back and forth at least ten times using the fanning technique. During needle actuation, the stylet was slowly withdrawn to fit the slow-pull technique. After each pass, the tissue was retrieved, and the stylet was inserted into the needle until the specimens were extruded through the needle tip onto a glass slide for visual inspection. MOSE was then performed by identifying a visible tissue core of ≥4 mm in length. A core tissue mixed with clots was acceptable. The total length of the tissue was measured by a ruler. Once a tissue core of at least 4 mm was obtained, the FNB was considered completed. The specimens were placed in 10% formalin for histological analysis. ROSE by a cytopathologist was not performed.

### 2.4. Tissue Processing for Histological Analysis

Core tissue was collected in 10% formalin solution for cell block preparation and histological examination. Formalin-fixed tissues were embedded in paraffin, and slides were made using standard technique [10]. The slides were stained with hematoxylin and eosin, and periodic acid-Schiff stains and were reviewed for histologic features. The specimens were assessed by an experienced pathologist specializing in gastrointestinal and pancreaticobiliary diseases who remained blind to patients’ history and laboratory results. Immunohistochemical staining was further performed on the cell blocks depending on the pathologist’s decision.

### 2.5. Definition of Histological Interpretation

The specimens were considered adequate if the acquired material provided sufficient tissue architecture for histological evaluation. The histological diagnosis was categorized as unsatisfactory, negative for malignancy, atypical, suspicious for malignancy, or positive for malignancy. A diagnosis of malignancy was made if the histological analysis was reported as being positive or suspicious for malignancy. Reports negative for malignancy and atypia were categorized as non-malignant. We did not include cases with inadequate specimens for post hoc analysis.

### 2.6. Criteria for Final Diagnosis

The final diagnoses were made based on one of the following criteria: (1) surgical pathology from the resected specimens, (2) histology from tissue obtained via EUS-FNB with or without ancillary studies or immunostaining, and (3) a minimum of 6 months of follow-up for clinical evaluation and interval imaging. Findings suggestive of malignancy during a 6-month follow-up included (1) new radiographic abnormalities such as regional or distant metastases, (2) direct invasion of the mass into vascular structures or adjacent organs, and (3) cancer-related mortality. The diagnosis of benign conditions required a minimum of 6 months of follow-up with resolution or stabilization of clinical symptoms and abnormal imaging.

### 2.7. Sample Size Estimation

We estimated that the sensitivity and specificity of EUS-FNB for diagnosing malignant causes of intraabdominal masses would be 90% and 100%, respectively [11]. Accounting for a 75% prevalence of malignancy and a 5% dropout rate, a minimum sample size of 100 was estimated to achieve a power of 0.80 with an alpha error of 0.05.

### 2.8. Statistical Analysis

The analyses were performed using SPSS version 16 (SPSS, Inc., Chicago, IL, USA). Continuous variables were presented as the mean ± standard deviation or as the median and interquartile range (IQR), and categorical variables were presented as a number and a percentage. The diagnostic performance of EUS-FNB was calculated as sensitivity, specificity, positive predictive value (PPV), and negative predictive value (NPV) with 95% confidence intervals (95% CI). The area under the receiver operating characteristics (AUROC) curve was constructed to assess accuracy. A logistic regression analysis was applied to identify factors influencing tissue adequacy. The data were presented as an odds ratio (OR) with a 95% CI. A p-value of less than 0.05 was considered statistically significant.

## 3. Results

### 3.1. Characteristics of the Study Population

The patient demographics and final diagnoses are shown in Table 1. One hundred patients with intra-abdominal mass lesions were enrolled during the study period. The mean age was 61.6 + 14.1 years (23 to 89 years), and 54% were male. Weight loss was the most common presentation, accounting for 58%, followed by abdominal pain (51%) and jaundice (27%). Surgery was performed in 23 patients. Ninety-five patients completed clinical and imaging follow-up at six months, and five patients died from underlying cancer diagnosis within six months of initial presentation. Seventy-eight patients had neoplasms, with pancreatic adenocarcinoma being the most common etiology, accounting for 27%. Other neoplastic lesions included mesenchymal tumor, metastasis, cholangiocarcinoma, neuroendocrine tumor, esophageal cancer, gastric cancer, gallbladder cancer, and lymphoma. Benign conditions were reactive inflammatory changes, chronic pancreatitis, and infections, including mycobacterium tuberculosis and cryptococcosis.

### 3.2. Endosonographic Findings

The endosonographic characteristics and EUS data are summarized in Table 2. The mean size of the masses was 3.9 ± 2.6 cm; 3% of lesions were <1cm, 18% were <2 cm, 25% were 2–3 cm, 23% were 3–4 cm, and 31% were larger than 4 cm. The most common site of masses was the pancreas (*n* = 41), followed by subepithelial lesions (*n* = 26) and intraabdominal lymph nodes (*n* = 19). The vast majority of the masses were hypoechoic (51%) and heteroechoic (46%). The median number of passes was 3 (IQR 1–5) for all sampled masses. All of the lesions were aspirated using a 22-gauge Franseen needle with the slow-pull technique with a reported technical success, defined by the completion of the tissue acquisition process until adequate tissue was obtained based on a MOSE, of 100% without needle malfunction.

### 3.3. Tissue Adequacy

The percentage of cases in which adequate specimens were obtained for histology was 89%. For histological grading, 57 out of 100 patients were positive for malignancy, 7 were suspicious for malignancy, 6 demonstrated atypia, and 19 were negative for malignancy, as shown in Table 3. Thirty-five patients required immunohistochemical staining for a definitive diagnosis, and the tissue was adequate for additional staining in all. Of the 11 patients with an inadequate specimen, 3 patients underwent surgery due to high clinical suspicion for malignancy and 8 patients had clinical and imaging follow-up for a minimum of 6 months. All of the patients who underwent surgery had malignancies, including cholangiocarcinoma (*n* = 2) and pancreatic cancer (*n* = 1). Eight of the remaining patients who had clinical and imaging follow-up further showcased reactive changes (*n* = 5), chronic pancreatitis (*n* = 2), and cholangiocarcinoma (*n* = 1).

In the univariate analysis, the size of the tumor and malignant diseases were significantly associated with a higher chance of obtaining adequate tissue (Table 4), whereas the type of cancer, location of the lesion, and the number of needle passes were not associated with tissue adequacy. In multivariate analysis, malignancy (OR 4.58, 95% CI, 1.15–18.2) remained an independent predictor of achieving tissue adequacy.

### 3.4. Diagnostic Performance of EUS-FNB with Histological Evaluation

The histological analysis obtained from the 22-gauge Franseen needles in solid lesions is demonstrated in Table 5. It provided excellent diagnostic performance in diagnosing malignancy, with an AUROC of 0.92. For pancreatic cancer, FNB had a sensitivity of 92.3%, a specificity of 100%, a PPV of 100%, an NPV of 97%, and an AUROC of 0.96. FNB with histological examination and immunohistochemistry staining was found to be highly sensitive for diagnosing mesenchymal cell tumors. The sensitivity was 100%, the specificity was 95.9%, the NPV was 100%, and the AUROC was 0.98. Furthermore, FNB was 100% sensitive and 100% specific in detecting metastatic cancers, underscoring the high diagnostic accuracy and relevance of this type of biopsy needle using the slow-pull technique. For benign lesions, FNB was outstanding for detecting tuberculous lymphadenitis, with a sensitivity of 100% (95% CI, 39.8–100), a specificity of 100% (95% CI, 92.9–100), a PPV of 100% (95% CI, 39.8–100), an NPV of 100% (95% CI, 92.9–100), and an AUROC of 1.00 (95% CI, 1.00–1.00).

### 3.5. Adverse Events

There were no procedure-related complications in all studied patients.

## 4. Discussions

The invention of FNB has resulted in a paradigm shift in EUS-guided tissue acquisition. EUS-FNB has emerged rapidly in clinical practice owing to its high technical success and promising diagnostic performance observed in prior studies. The main novel biopsy needle designs include Procore (reversed bevel), Fork-tip, Franseen, and 20-gauge Procore (forward bevel). The present study prospectively evaluated the diagnostic performance of 22-gauge Franseen needles with histological analysis in the absence of ROSE to diagnose intraabdominal solid masses, including pancreatic and extrapancreatic lesions. Additionally, we uniformly performed the slow-pull technique and applied MOSE in the specimen evaluation. The results showed that the technical success was 100%, and the tissue adequacy for histological analysis was 89%. The EUS-FNB without ROSE was excellent in diagnosing pancreatic cancer, mesenchymal tumors, and metastases with AUROCs of 0.96, 0.98, and 1.00, respectively.

Both needle size and design are important factors that influence needle performance and have been widely studied. Our findings are consistent with those of earlier investigations evaluating the diagnostic accuracy of 22-gauge Franseen needles in the presence of ROSE [1,2,3,7]. Mita et al. found that, after the first pass, the diagnostic accuracy in identifying cancer of solid masses was 93%, and after the three cumulative passes, the accuracy increased to 96%. The authors utilized 20 mL negative pressure suction without ROSE or MOSE in their investigation [6]. In addition, 25-gauge Franseen needles were also evaluated. A prospective multicenter trial of 100 patients found that, after utilizing 20 mL negative pressure suction without ROSE, the 25-gauge needles obtained core tissue with a 95% acquisition rate [5]. When compared with a historical series, the 20-gauge Procore showed a clear trend toward better performance, without the added requirement of ROSE [12,13]. Clinical studies evaluating the size of the needle for its ability to improve diagnostic information have been conducted, with the main findings being that the 20-gauge FNB needle is superior to the 25-gauge FNB needle for retrieving tissue samples [14]. A multicenter group trial comparing the 20-gauge Procore needles versus the 22-gauge Acquire needles showed that histologic diagnoses were achieved in 40/60 20-gauge Procore and 52/60 22-gauge Acquire needles. The length of tissue samples was better in the 22-gauge Acquire needles, with a greater mean surface area of preserved tissue [15]. Furthermore, a comparison of Franseen and Fork-tip biopsy needles showed that both types of needles provided comparable tissue adequacy [16,17,18], with one study confirming Franseen needle’s superiority in providing better diagnostic accuracy (96% vs. 92.4%, *p*-value < 0.001) [16].

Additionally, much debate exists regarding the slow-pull versus standard suction technique in EUS-FNB. A prospective randomized trial comparing these two aspiration techniques with the 20-gauge Procore needles was also conducted and showed that the slow-pull and suction techniques are comparable in terms of sensitivity, accuracy, and blood contamination in pancreatic lesions [19]. Conversely, another study demonstrated that the slow-pull technique was associated with higher diagnostic accuracy [20]. For Franseen needles, Ishigaki et al. demonstrated that suction was associated with reduced accuracy for pancreatic lesions [7]. In our present study, we assessed the slow-pull technique and reported excellent diagnostic performance in detecting malignancy and infections without significant blood contamination. It remains to be explored whether different aspiration techniques are required for different needle types—a foundation for future studies to come.

ROSE has traditionally been used to assess tissue adequacy. MOSE has recently been introduced to clinical practice, but the method has yet to be standardized. A macroscopic visible core of ≥4 mm in length is associated with a higher diagnostic yield for cytological and histological evaluation [21]. Chong et al. demonstrated that MOSE, using the cut-off of macroscopic visible core ≥4 mm in length, provides a similar yield to conventional EUS-FNA without ROSE but requires fewer passes [22]. With the Franseen needles, visualization of the core tissue by MOSE (50/54 patients) was well correlated with pathologically confirmed core (47/50 patients) [8]. In our study, MOSE was used instead of ROSE to determine tissue adequacy. We elected to use the cut-off of a visible core tissue length of 4 mm based on Iwashita et al. [21]. However, we noted a discrepancy between MOSE and the cytologist’s assessment. Despite applying MOSE, the pathologist reported that 89% of patients had adequate tissue for histological evaluation. The limitations of macroscopic assessment included the presence of fibrous tissue, neighboring stomach and duodenal mucosa, and blood contamination disrupting accurate interpretation. Existing studies are emerging to confirm that EUS-FNB requires a lower number of needle passes to achieve proper diagnostic potential when compared with FNA [23,24].

When using FNA needles, the specimens should be prepared for both cytological and histological analysis. Nonetheless, tissue handling methods have not been standardized, and clinical practice varies. With FNB needles, the specimens can be placed in formalin and processed for histology examination alone, obviating the need for ROSE and tissue processing for cytology. Chen et al. conducted a non-inferiority study comparing the performance of EUS-FNB without ROSE versus EUS-FNA with ROSE in pancreatic lesions. In this study, a variety of aspiration techniques, including standard negative suction, wet suction, and slow-pull, have been used with 22-gauge and 25-gauge Fork-tip needles. The study found that EUS-FNB was comparable to EUS-FNA with ROSE in terms of accuracy, sensitivity, and specificity. The EUS-FNB required fewer passes and took less time, although it remained slightly more expensive [9]. The utility of ROSE in FNB was further examined in a non-inferiority trial comparing the diagnostic accuracy of FNB alone and FNB with ROSE. This study demonstrated comparable accuracy in both arms, with a higher tissue core rate in EUS-FNB without ROSE [25]. A head-to-head study design comparing the use of ROSE vs. FNB without ROSE may give a better perspective about the additional effects of ROSE and cytological examination on the diagnostic yield when EUS-FNB with MOSE is employed.

The study’s strengths are that (1) all patients were followed until the end of the study without loss to follow-up or missing data; (2) the endoscopic procedure and MOSE were performed by a dedicated, experienced endosonographer, eliminating variation in techniques and interobserver disagreement; and (3) the histological specimen was examined by a dedicated pathologist blinded to clinical data, minimizing bias and interobserver disagreement. Nonetheless, the study’s limitations were (1) the lack of a comparator, (2) procedures performed by an operator using one type and size of the needle in a single center, and (3) the correlation between tissue adequacy and the result of MOSE per pass was not assessed.

## 5. Conclusions

Our findings support the notion that FNB alone provides excellent diagnostic accuracy and that ROSE with cytological examination may not be routinely indicated. MOSE should be implemented in real-world practice, and ROSE can be obviated when EUS-FNB is employed. The advantage of using FNB to better characterize immunologic and molecular detail through next-generation molecular profiling is already underway and will continue to underscore FNB’s importance in EUS-guided tissue acquisition.

## Figures and Tables

**Table 1 jcm-11-01051-t001:** Patient characteristics of the study population.

Parameter	Value
Age, year (mean ± SD)	61.6 ± 14.1
Sex, *n* (%)	
Male	54 (54)
Female	46 (46)
Clinical manifestation, *n* (%)	
Weight loss	58 (58)
Abdominal pain	51 (51)
Jaundice	27 (27)
Palpable abdominal mass	11 (11)
Anemia	4 (4)
Elevated liver enzymes	1 (1)
Abnormal imaging	21 (21)
Definite diagnosis, *n* (%)	
Malignancy	78 (78)
Pancreatic adenocarcinoma	27 (27)
Mesenchymal tumor	17 (17)
Metastasis	14 (14)
Cholangiocarcinoma	8 (8)
Neuroendocrine tumor	4 (4)
Esophageal cancer	2 (2)
Gastric cancer	2 (2)
Gallbladder cancer	2 (2)
Lymphoma	2 (2)
Inflammation or reactive changes	13 (13)
Infections *	5 (5)
Others **	4 (4)

NOTE. Data are presented as mean ± standard deviation or the number (%) of patients with a condition. * Infections, mycobacterium tuberculosis = 4; cryptococcosis = 1. ** Others, chronic pancreatitis = 1; low-grade schwannoma = 1; immunoglobulin G4-related disease = 1; cavernous hemangioma = 1. SD, standard deviation.

**Table 2 jcm-11-01051-t002:** Endosonographic data and features of intraabdominal masses.

Characteristics	
Technique, *n* (%)	
Slow pull technique	100 (100)
Number of the needle passes	
1	3 (3)
2	37 (37)
3	51 (51%)
4	8 (8%)
5	1 (1%)
Median pass number (IQR)	3 (1–5)
Size (cm) (mean ± SD)	3.99 ± 2.62
Location, *n* (%)	
Pancreas	41 (41)
Head	28 (28)
Body	8 (8)
Tail	5 (5)
Subepithelial lesions	26 (26)
Esophagus	8 (8)
Stomach	16 (16)
Second part duodenum	2 (2)
Abdominal lymph nodes	19 (19)
Liver	7 (7)
Retroperitoneum	1 (1)
Others *	6 (6)
Echogenicity, *n* (%)	
Hypoechoic	51 (51)
Heteroechoic	46 (46)
Hyperechoic	2 (2)
Isoechoic	1 (1)
Complications	0

NOTE. Data are presented as mean ± standard deviation, median (IQR), or the number (%) of patients with a condition. * Others; CBD = 2, gallbladder = 4. IQR, interquartile range; SD, standard deviation.

**Table 3 jcm-11-01051-t003:** Histological assessment of sampling specimens from EUS-FNB.

Parameters	No N (%) (%)
Tissue adequacy	
Yes	89 (89)
No	11 (11)
Grading	
Negative for malignancy	19 (19)
Atypia	6 (6)
Suspicious for malignancy	7 (7)
Positive for malignancy	57 (57)

NOTE. Data are presented as the number (%) of patients with a condition.

**Table 4 jcm-11-01051-t004:** Factors associated with tissue adequacy.

	Univariate Analysis		Multivariate Analysis	
Variables	Unadjusted OR (95% CI)	*p*-Value	Adjusted OR (95% CI)	*p*-Value
Malignancy	6.54 (1.73–24.7)	0.006	4.58 (1.15–18.2)	0.031
Pancreatic cancer	4.06 (0.49–33.4)	0.192		
Hepatobiliary cancer	0.30 (0.07–1.32)	0.111		
Subepithelial lesion	3.85 (0.47–31.6)	0.210		
Lymphadenopathy	0.62 (0.15–2.59)	0.513		
Size of the lesion	1.73 (1.03–2.90)	0.037	1.53 (0.92–2.54)	0.102
Hypoechogenicity	1.37 (0.39–4.82)	0.622		
Heterogeneous echogenicity	0.64 (0.18–2.24)	0.483		
Number of the needle passes	0.55 (0.24–1.28)	0.167		

NOTE. The multivariate model includes malignancy and size of the lesion, which were significant in the univariate analysis. CI, confidence interval; OR, odds ratio.

**Table 5 jcm-11-01051-t005:** Diagnostic performance of EUS-FNB for solid neoplasms.

Diagnosis	AUROC	Sensitivity (95% CI)	Specificity (95% CI)	PPV (95% CI)	NPV (95% CI)
Malignancy	0.92 (0.84–1.00)	95.8 (88.1–99.1)	88.9 (65.3–98.6)	97.1 (90.1–99.7)	84.2 (60.4–96.6)
Pancreatic cancer	0.96 (0.91–1.00)	92.3 (74.9–99.1)	100 (94.4–100)	100 (85.8–100)	97 (89.5–99.6)
Primary hepatobiliary malignancy	0.89 (0.74–1.00)	77.8 (40–97.2)	100 (95.5–100)	100 (59–100)	97.6 (91.6–99.7)
Mesenchymal tumor	0.98 (0.96–1.00)	100 (79.4–100)	95.9 (88.6–99.2)	84.2 (60.4–96.6)	100 (94.9–100)
Metastatic cancer	1.00 (1.00–1.00)	100 (76.8–100)	100 (95.3–100)	100 (76.8–100)	100 (95.3–100)

AUROC, area under the receiver operating characteristics curve; PPV, positive predictive value; NPV, negative predictive value; CI, confidence interval.

## Data Availability

All the data are included in the manuscript.

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
