# Peer review of "Endoscopic Ultrasound-Guided Fine-Needle Biopsy Using 22G Franseen Needles without Rapid On-Site Evaluation for Diagnosis of Intraabdominal Masses"

_jcm, 2022, doi:10.3390/jcm11041051_

Round 1
Reviewer 1 Report
This is a prospective study that evaluated the Franseen needle during EUS-FNA for solid lesions. This study contains informative results, although the reviewer has several comments for this article.
Comment: Abstract
This study included 100 pts, 78 with neoplasms and 23 with benign. A total number cannot be 100. Please correct it.
Comment: Methods
FNA was continued until a visible tissue of >= 4mm in length was collected. What was the meaning of “4mm” and was there any rule for the maximum number of punctures?
Comment: Sample size
The sample size calculation cannot be made based on the given information. Please provide the threshold accuracy.
Comment: Multivariate analysis
OR was significantly higher with malignancy. Did the analysis evaluate tissue adequacy? Then, higher OR means we can easily obtain adequate tissue. Please clearly describe this point. Also, univariate analysis showed the significance of the size of the lesion. The authors should include this factor in the multivariate analysis. The authors included “number of the needle passes” as one of the variables. The, OR might be unit OR? Please clearly state this point as well.
Comment: ROC
Usually, ROC is used with a combination of continuous and dichotomous variables to evaluate the optimal cut-off level of the continuous variable or efficacy of the test with AUC. However, the reviewer is not sure which continuous variable was used in this analysis and the meaning of AUC. Please discuss statistical analysis with your statistician.
Author Response
Reviewer 1
This is a prospective study that evaluated the Franseen needle during EUS-FNA for solid lesions. This study contains informative results, although the reviewer has several comments for this article.
Response: We thank the reviewer for taking the time to review our manuscript and providing helpful comments. We have revised our manuscript to address the reviewer’s comments, and a point-by-point response to the specific questions is provided. Also, the manuscript has been English proofread by a native English speaker.
Comment: Abstract
This study included 100 pts, 78 with neoplasms and 23 with benign. The total number cannot be 100. Please correct it.
Response: A correction has been made. “Seventy-eight patients had neoplasms, and 22 patients had benign diseases.” (page 1, line 25)
Comment: Methods
FNA was continued until a visible tissue of >= 4mm in length was collected. What was the meaning of “4mm,” and was there any rule for the maximum number of punctures?
Response: The macroscopic on-site evaluation (MOSE) of the aspirates has been introduced; however, the method has not been standardized. Iwashita et al. reported that the optimal cutoff length for a macroscopic visible core (MVC) was 4 mm. An MVC of ≥ 4 mm in length is associated with higher tissue adequacy and diagnostic yield. (Iwashita T, Yasuda I, Mukai T, Doi S, Nakashima M, Uemura S, et al. Macroscopic on-site quality evaluation of biopsy specimens to improve the diagnostic accuracy during EUS-guided FNA using a 19-gauge needle for solid lesions: a single-center prospective pilot study (MOSE study). Gastrointestinal Endoscopy. 2015;81(1):177-85). Therefore, we elected to use the tissue core cutoff of 4 mm in length to determine tissue adequacy. The detailed explanation of how MOSE was performed in our study has been added to the EUS-FNB technique under the Materials & Methods section (page 3, lines 109-112) to improve the clarity of the manuscript. Also, we added the reason why we have chosen the cut-off of 4 mm in the discussion section (page 8, lines 295-296).
In terms of the maximum number of punctures, there is no definite rule on when to stop puncturing. However, Jhala et al. demonstrated that 90% of adequate samples were obtained within 6 passes, after which there was only a slight increase in obtaining an adequate sample (Jhala NC, Jhala D, Eltoum I, Vickers SM, Wilcox CM, Chhieng DC, et al. Endoscopic ultrasound-guided fine-needle aspiration biopsy: A powerful tool to obtain samples from small lesions. Cancer cytopathology. 2004;102(4):239-46). Thus, it is our common practice to puncture with a maximum of 6-7 passes.
Comment: Sample size
The sample size calculation cannot be made based on the given information. Please provide the threshold accuracy.
Response: Additional information, including the threshold accuracy, has been added as the reviewer suggested (page 3-4, lines 142-147). We estimated that the sensitivity and specificity of EUS-FNB for diagnosing malignant causes of intraabdominal masses would be 90% and 100%, respectively. With accounting for a 75% prevalence of malignancy and a 5% dropout rate, a minimum sample size of 100 was estimated to achieve a power of 0.80 with an alpha error of 0.05.
Statistical analysis
Comment: Multivariate analysis
OR was significantly higher with malignancy. Did the analysis evaluate tissue adequacy? Then, higher OR means we can easily obtain adequate tissue. Please clearly describe this point. Also, univariate analysis showed the significance of the size of the lesion. The authors should include this factor in the multivariate analysis. The authors included “number of the needle passes” as one of the variables. The, OR might be unit OR? Please clearly state this point as well.
Response: Yes, the multivariate analysis was performed to identify factors predictive of tissue adequacy. Therefore, the higher the OR is, the more chance to obtain adequate tissue. We have clarified this issue as the reviewer kindly suggested. Changes have been made on page 6, lines 208-212 under the tissue adequacy section.
Regarding the univariate analysis, we added the size of the lesion in the multivariate analysis as suggested. However, this variable was not significant for tissue adequacy in multivariate analysis (p=0.102). The change has been made to Table 4. Factors Associated with Tissue Adequacy on page 6.
The odds ratio for the number of the needle passes means that as one needle pass increases, the likelihood of obtaining adequate tissue decreases.
Comment: ROC
Usually, ROC is used with a combination of continuous and dichotomous variables to evaluate the optimal cutoff level of the continuous variable or efficacy of the test with AUC. However, the reviewer is not sure which continuous variable was used in this analysis and the meaning of AUC. Please discuss statistical analysis with your statistician.
Response: The area under the receiver operating characteristic (AUROC) can be used to calculate how accurately a diagnostic test can distinguish between those with and without the disease. Diagnostic tests employ continuous measurement scales, resulting in a score range with many possible cutoff values and, therefore, multiple points on an AUROC plot, with linear interpolation between the points (the plot tends to a curve as the number of points approaches infinity).
In the AUROC space, there will be n – 1 point for a categorical classifier with n thresholds. Because a binary classifier (i.e., the presence or absence of malignant disease assessed by FNB in this study) has only a single fixed threshold. It is represented by a receiver operating characteristic (ROC) dot rather than a ROC plot. In this circumstance, test accuracy (AUROC) for identifying the cause of solid lesions is calculated using the area of a triangle rather than the area under a curve. While it can be difficult for researchers to determine the optimal Sensitivity and Specificity values for their diagnostic test to be considered accurate, the AUROC result takes both values into account to produce a single value representing the overall diagnostic accuracy of the test, interpreted on an externally validated scale.
Reviewer 2 Report
In this paper, the authors investigated the histological diagnostic performance of the 22-gauge Franseen needle using slow-pull aspiration with MOSE.
I consider that the manuscript is a well-organized and informative study for the readers. However, there are several concerns in this article.
Major comments
- Current study showed median pass number of FNB was 3. This result seems to be more than other studies about FNB with MOSE. How did the authors determine the visible tissue and how did authors measure the visible tissue? Is there any possibility that slow pull method is not suitable for MOSE because of less visible tissue?
- The tissue adequacy was reported 89%. How is the relationship between tissue adequacy and the result of MOSE of per pass?
- Authors mentioned not only ROSE but also cytological examination may not be routinely indicated in EUS-FNB with MOSE. Is there any data support this suggestion in current study or previous study? Is there no additional effect for diagnostic yield to add cytological examination for histological examination in EUS-FNB with MOSE?
Author Response
Reviewer 2
Comments and Suggestions for Authors
In this paper, the authors investigated the histological diagnostic performance of the 22-gauge Franseen needle using slow-pull aspiration with MOSE.
I consider that the manuscript is a well-organized and informative study for the readers. However, there are several concerns in this article.
Response: We thank the reviewer for taking the time to review our manuscript and providing helpful comments. We have revised our manuscript to address the reviewer’s comments, and a point-by-point response to the specific questions is provided.
Major comments
1.Current study showed median pass number of FNB was 3. This result seems to be more than other studies about FNB with MOSE. How did the authors determine the visible tissue and how did authors measure the visible tissue? Is there any possibility that slow pull method is not suitable for MOSE because of less visible tissue?
Response: The macroscopic on-site evaluation (MOSE) of the aspirates has been introduced; however, the method has not been standardized. Iwashita et al. reported that the optimal cutoff length for a macroscopic visible core (MVC) was 4 mm. An MVC of ≥ 4 mm in length is associated with higher tissue adequacy and diagnostic yield. (Iwashita T, Yasuda I, Mukai T, Doi S, Nakashima M, Uemura S, et al. Macroscopic on-site quality evaluation of biopsy specimens to improve the diagnostic accuracy during EUS-guided FNA using a 19-gauge needle for solid lesions: a single-center prospective pilot study (MOSE study). Gastrointestinal Endoscopy. 2015;81(1):177-85). Therefore, we elected to use the tissue core cutoff of 4 mm in length to determine tissue adequacy. The detailed explanation of how MOSE was performed in our study has been added to the EUS-FNB technique under the Materials & Methods section (page 3, lines 109-112) to improve the clarity of the manuscript. Briefly, the tissue was retrieved after each pass. The stylet was inserted into the needle until the specimens were extruded through the needle tip onto a glass slide for visual inspection. MOSE was then performed by identifying a visible tissue core of ≥ 4 mm in length. A core tissue mixed with clots was acceptable. The total length of tissue was measured by a ruler. Once a tissue core of at least 4 mm was obtained, the FNB was considered completed. The specimens were placed in 10% formalin for histological analysis. Also, we added the reason why we have chosen the cut-off of 4 mm in the discussion section (page 8, lines 295-296).
The reviewer’s comment on the utility slow pull method in MOSE is very well taken. Iwashita et al. and Chong et al. (Chong CCN, Lakhtakia S, Nguyen N, Hara K, Chan WK, Puri R, et al. Endoscopic ultrasound-guided tissue acquisition with or without macroscopic onsite evaluation: randomized controlled trial. Endoscopy. 2020;52(10):856-63) assessed the role of MOSE in EUS-FNA with 19 G FNA needles and suction technique. However, a recent study by Gaia et al. (Gaia S, Rizza S, Bruno M, Ribadone DG, Maletta F, Sacco M, et al. Impact of Macroscopic On-site Evaluation (MOSE) on accuracy of endoscopic ultrasound-guided fine needle biopsy (EUS-FNB) of pancreatic and extrapancreatic solid lesions: A prospective study. Diagnostics. 2022) assessed the utility of MOSE in EUS-FNB using 22G/25G Franseen needles, and both slow pull and suction techniques were used. The results showed that MOSE provided accuracy of 86.8%, and adequacy of 97.4%, suggesting that the slow pull method can be used for MOSE. It is worth mentioning that the classification for MOSE in these studies is different. Gaia et al. required only 2 mm of visible whitish core tissue.
- The tissue adequacy was reported 89%. How is the relationship between tissue adequacy and the result of MOSE of per pass?
Response: This study was not designed to assess the tissue adequacy and MOSE result per pass. Therefore, our study could not draw any conclusion related to this matter. We addressed this issue by stating it as a limitation in the discussion section (page 9, lines 325-326). However, the study by Gaia et al. assessed the tissue adequacy and accuracy per pass. They reported that the third pass increased the diagnostic accuracy of the first two passes from 82.6% to 87% (p = 0.87), but the difference was not significant.
- Authors mentioned not only ROSE but also cytological examination may not be routinely indicated in EUS-FNB with MOSE. Is there any data support this suggestion in current study or previous study? Is there no additional effect for diagnostic yield to add cytological examination for histological examination in EUS-FNB with MOSE?
Response: There are two large non-inferiority, randomized controlled trials which aimed to assess the efficacy of ROSE and cytological examination when EUS-FNB with MOSE is employed. Chen et al. conducted a non-inferiority study comparing the performance of EUS-FNB without ROSE versus EUS-FNA with ROSE in pancreatic lesions. The study found that EUS-FNB was comparable to EUS-FNA with ROSE in accuracy, sensitivity, and specificity. (Chen Y-I, Chatterjee A, Berger R, et al. Endoscopic ultrasound (EUS)-guided fine needle biopsy alone vs. EUS-guided fine needle aspiration with rapid onsite evaluation in pancreatic lesions: a multicenter randomized trial. Endoscopy. 2021)
The benefit of ROSE in FNB was further examined in a study comparing the diagnostic accuracy of FNB alone and FNB with ROSE. This non-inferiority trial demonstrated comparable accuracy in both arms with a higher tissue core rate in EUS-FNB without ROSE. (Crinò SF, Mitri RD, Nguyen NQ, et al. Endoscopic Ultrasound–guided Fine-needle Biopsy With or Without Rapid On-site Evaluation for Diagnosis of Solid Pancreatic Lesions: A Randomized Controlled Non-Inferiority Trial. Gastroenterology 2021;161(3):899-909)
Based on these two studies, we conclude that EUS-FNB without ROSE is not inferior to EUS-FNB with ROSE and EUS-FNA with ROSE. However, larger prospective head-to-head comparison studies are needed to determine whether cytological examination adds diagnostic value to EUS-FNB with MOSE. This notion has been added to the discussion section (page 9, lines 319-322)
The reviewer’s comment is well taken that our study was not designed to compare the benefit of ROSE and no ROSE in light of EUS-FNB with MOSE; thus, we could not conclude that EUS-FNB with MOSE is superior to EUS-FNB with ROSE based on the results of this study. We have removed the following sentence from the discussion- “The present study adds to the collection of experiences demonstrating the superiority of FNB-based techniques in improving tissue sampling for histologic evaluation without added requirement of ROSE.”(page 8, lines 314-316)
Reviewer 3 Report
First of all, congratulations to the investigators for conducting this study and successfully. Here are a few suggestions that your team can consider to improve the quality of the manuscript.
2. Please make the title more concise.
25. Please mention metastatic tumors not just metastases.
27. Rephrase this sentence.
31. replace with metastatic tumors
36. Consider Pancreatic cancer as a keyword, will likely give more citations if published.
42. Primarily used to biopsy strictures adjacent to upper GI tract.
59. Replace shows with showed
64. Replace ‘have’ with ‘are’
66. Technique should be singular
89. Experienced endoscopist who previously had performed more than 2000 cases…
92. Please state in plural, since multiple procedures were performed.
93. Please state how the length of the specimen was determined.
112. Add the word ‘stains’ in the sentence
114. Perhaps consider mentioning that the pathologist was also blinded to the knowledge of whether ROSE was performed, if this was not feasible then please mention that in the methods.
138. alpha error
175. Rephrase as: ‘median number of passes’
177. What do the authors refer to as technical success here?
178. Since all the lesions are performed during upper GI EUS, the authors should mention this in their introduction and methods that the study was limited for upper GI EUS exams.
216. Please state the number of cases of TB lymphadenitis diagnosed, I suspect this is a small number, which can give rise to a bias in reporting and the readers need to be aware of this fact.
244. replace with: 20 mL
252. Rephrase the sentence
269. I am glad the authors state that studied may need to be conducted with other needles, since this study is not generalizable for using needles of other sizes and different manufacturers.
272. Replace ‘has’ with ‘is’
273. Why did the authors chose to select 4 mm as a cut off? May consider detailing the reasoning in methods.
274. Other limitations to consider: single center, one operator, only one type of needle used, only one size of needle used.
The authors could mention that a head-to-head study design comparing the use of ROSE vs FNB without ROSE may give a better perspective about utility of ROSE.
Author Response
Reviewer 3
Comments and Suggestions for Authors
First of all, congratulations to the investigators for conducting this study and successfully. Here are a few suggestions that your team can consider to improve the quality of the manuscript.
Response: We are grateful for the reviewer’s input which helps improve the quality of our manuscript. Your time is greatly appreciated.
- Please make the title more concise.
Response: We have changed the title to “Endoscopic Ultrasound-Guided Fine Needle Biopsy Using 22G Franseen Needles without Rapid On-Site Evaluation For Diagnosis of Intraabdominal Masses”
- Please mention metastatic tumors not just metastases.
Response: The change has been made as suggested.
- Rephrase this sentence.
Response: The sentence has been rephrased as suggested. The revised sentence is “Malignancy increased the odds of obtaining adequate tissue (OR 5.53, 95% CI, 1.36-22.5).”
- replace with metastatic tumors
Response: The change has been made as suggested.
- Consider Pancreatic cancer as a keyword, will likely give more citations if published.
Response: Pancreatic cancer was added to the keyword list as suggested.
- Primarily used to biopsy strictures adjacent to upper GI tract.
Response: The change has been made as suggested.
- Replace shows with showed
Response: The change has been made as suggested.
- Replace ‘have’ with ‘are’
Response: The change has been made as suggested.
- Technique should be singular
Response: We changed “techniques” to “technique” as suggested.
- Experienced endoscopist who previously had performed more than 2000 cases…
Response: The change has been made as suggested.
- Please state in plural, since multiple procedures were performed.
Response: “The lesion was accessed….” has been changed to “The lesions were accessed…”
- Please state how the length of the specimen was determined.
Response: The length of the specimen was measured by a ruler. We added a detailed explanation of how MOSE was performed to this section as well.
- Add the word ‘stains’ in the sentence
Response: The change has been made as suggested.
- Perhaps consider mentioning that the pathologist was also blinded to the knowledge of whether ROSE was performed, if this was not feasible then please mention that in the methods.
Response: In our study, the pathologist was blinded to the patients’ clinical data but was not blinded to the presence of ROSE. Unfortunately, we could not make any change on this issue.
- alpha error
Response: The change has been made as suggested.
- Rephrase as: ‘median number of passes’
Response: The change has been made to “the median number of passes” as suggested.
- What do the authors refer to as technical success here?
Response: The technical success means the completion of the tissue acquisition process until adequate tissue was obtained based on the macroscopic on-site evaluation. The definition of technical success was added to the text.
- Since all the lesions are performed during upper GI EUS, the authors should mention this in their introduction and methods that the study was limited for upper GI EUS exams.
Response: We specified in the introduction and method that the EUS was limited to upper GI exams as suggested. (page 2, line 71 and line 83)
- Please state the number of cases of TB lymphadenitis diagnosed, I suspect this is a small number, which can give rise to a bias in reporting and the readers need to be aware of this fact.
Response: The reviewer’s comment is well taken. The number of cases of TB and cryptococcosis was added to the Table 1 footnote.
- replace with: 20 mL
Response: The change has been made as suggested.
- Rephrase the sentence
Response: The sentence has been rephrased. The revision is ‘Clinical studies evaluating the size of the needle for its ability to improve diagnostic information have been conducted, with the main findings being that the 20-gauge FNB needle is superior to the 25-gauge FNB needle for retrieving tissue samples’
- I am glad the authors state that studied may need to be conducted with other needles, since this study is not generalizable for using needles of other sizes and different manufacturers.
Response: We are thankful for the reviewer’s comment.
- Replace ‘has’ with ‘is’
Response: The change has been made as suggested.
- Why did the authors chose to select 4 mm as a cut off? May consider detailing the reasoning in methods.
Response: The macroscopic on-site evaluation (MOSE) of the aspirates has been introduced; however, the method has not been standardized. Iwashita et al. reported that the optimal cutoff length for a macroscopic visible core (MVC) was 4 mm. An MVC of ≥ 4 mm in length is associated with higher diagnostic yield. (Iwashita T, Yasuda I, Mukai T, Doi S, Nakashima M, Uemura S, et al. Macroscopic on-site quality evaluation of biopsy specimens to improve the diagnostic accuracy during EUS-guided FNA using a 19-gauge needle for solid lesions: a single-center prospective pilot study (MOSE study). Gastrointestinal Endoscopy. 2015;81(1):177-85). Therefore, we elected to use the tissue core cutoff of 4 mm in length to determine tissue adequacy. We discussed our reason for choosing this cut-off and added the detail of how MOSE was performed as suggested. Changes have been made in the methods and discussion section.
- Other limitations to consider: single center, one operator, only one type of needle used, only one size of needle used.
Response: Changes have been made as suggested.
The authors could mention that a head-to-head study design comparing the use of ROSE vs FNB without ROSE may give a better perspective about utility of ROSE.
Response: Changes have been made to the discussion as suggested.
Reviewer 4 Report
Nice paper, well documented and conducted with high scientific relevance in the field of ecoendoscopy. The references up dated.
Maybe correlations with lesions elastography will be interesting.
Author Response
Nice paper, well documented and conducted with high scientific relevance in the field of ecoendoscopy. The references up dated.
Maybe correlations with lesions elastography will be interesting.
Response: We are thankful for the reviewer’s comments. Your time reviewing our manuscript is much appreciated. This study was not designed to compare the findings with elastography. Thus, we are not able to provide the reviewer with more information related to elastography.
Round 2
Reviewer 1 Report
The authors well revised the manuscript. I do not have any further comments.
Reviewer 2 Report
Thank you for your revision. I consider that the manuscript is revised appropriately. I have no more suggestions.